# Autoimmunity against Nucleus Ambiguous Is Putatively Possible in Both Long-COVID-19 and Vaccinated Subjects: Scientific Evidence and Working Hypothesis

**DOI:** 10.3390/biology13060359

**Published:** 2024-05-21

**Authors:** Silvestro Ennio D’Anna, Alessandra Maria Vitale, Giuseppa D’Amico, Celeste Caruso Bavisotto, Pasquale Ambrosino, Francesco Cappello, Mauro Maniscalco, Antonella Marino Gammazza

**Affiliations:** 1Istituti Clinici Scientifici Maugeri IRCCS, 27100 Pavia, Italy; silvestro.danna@icsmaugeri.it (S.E.D.); pasquale.ambrosino@icsmaugeri.it (P.A.); 2Department of Biomedicine, Neuroscience and Advanced Diagnostics (BIND), University of Palermo, 90127 Palermo, Italy; alessandramaria.vitale@unipa.it (A.M.V.); giuseppa.damico01@unipa.it (G.D.); celeste.carusobavisotto@unipa.it (C.C.B.); francesco.cappello@unipa.it (F.C.); 3Euro-Mediterranean Institute of Science and Technology (IEMEST), 90139 Palermo, Italy; 4National Biodiversity Future Center (NBFC), Piazza Marina 61, 90133 Palermo, Italy; 5Department of Clinical Medicine and Surgery, University of Naples “Federico II”, 80131 Naples, Italy

**Keywords:** SARS-CoV-2, long-COVID syndrome, postural orthostatic tachycardia syndrome, autoimmunity, molecular mimicry, vaccination

## Abstract

**Simple Summary:**

COVID-19, with persistent and new onset of symptoms, such as fatigue, post-exertional malaise, and cognitive dysfunction that impact everyday functioning, is referred to as long-COVID under the general category of post-acute sequelae of the SARS-CoV-2 infection (PASC). It includes a wide range of signs and symptoms that can last weeks, months, or even years after infection, most of which are attributable to dysfunctions of the neurovegetative system. The causative mechanisms are still unknown, but autoimmunity and the production of autoantibodies targeting self-antigens via the molecular mimicry phenomenon seem to have a role. Here we evaluated the presence of autoantibodies against two proteins of vagal nuclei sharing a peptide with SARS-CoV-2 spike glycoprotein in sera from ongoing symptomatic COVID-19 patients (long-COVID) with cardiorespiratory symptoms, subjects vaccinated without a history of SARS-CoV-2 infection, and subjects not vaccinated without a history of SARS-CoV-2. Putative autoantibodies are present in both long-COVID-19 and vaccinated groups, suggesting that both viral infection and vaccination may trigger autoreactivity. However, the presence of autoantibodies is not sufficient for triggering autoimmunity, and other predisposing conditions must co-occur. Therefore, it is necessary to run further investigations to clarify the complex mechanisms involved in the development of long-COVID, providing knowledge which may offer further information for the prevention and treatment of the disease.

**Abstract:**

As reported by the World Health Organization (WHO), about 10–20% of people have experienced mid- to long-term effects following SARS-CoV-2 infection, collectively referred to as post-COVID-19 condition or long-COVID, including some neurovegetative symptoms. Numerous findings have suggested that the onset of these neurovegetative symptoms upon viral infection may be caused by the production of autoantibodies through molecular mimicry phenomena. Accordingly, we had previously demonstrated that 22 of the human proteins sharing putatively immunogenic peptides with SARS-CoV-2 proteins are expressed in the dorsal motor nucleus and nucleus ambiguous. Therefore, if molecular mimicry occurs following severe forms of COVID-19, there could be transitory or permanent damage in some vagal structures, resulting in a lower vagal tone and all the related clinical signs. We investigated the presence of autoantibodies against two proteins of vagal nuclei sharing a peptide with SARS-CoV-2 spike glycoprotein using an immunoassay test on blood obtained from patients with cardiorespiratory symptoms in patients affected by ongoing symptomatic COVID-19 (long-COVID), subjects vaccinated without a history of SARS-CoV-2 infection, and subjects not vaccinated without a history of SARS-CoV-2 infection. Interestingly, putative autoantibodies were present in both long-COVID-19 and vaccinated groups, opening interesting questions about pathogenic mechanisms of the disease.

## 1. Introduction

As is known, some patients who have contracted SARS-CoV-2 infection can experience mid- to long-term effects, including cardiovascular, respiratory, and nervous effects such as pain on breathing, palpitations, variations in heart rate, and chest pain. In addition to clinical symptoms, people report increased absence or reduced performance in their education, work, or training. All these symptoms have been collectively referred to as long-COVID syndrome [1]. According to the last (2024) updated NICE guidelines, the following clinical case definitions are useful to identify and diagnose the long-term effects of COVID-19: (i) acute COVID-19 signs and symptoms of COVID-19 persist for up to 4 weeks; (ii) ongoing symptomatic COVID-19 signs and symptoms of the disease from 4 weeks up to 12 weeks; (iii) post-COVID-19 syndrome signs and symptoms continue for more than 12 weeks and are not explained by an alternative diagnosis. In addition to the clinical case definitions, the term ‘long-COVID’ is commonly used to describe signs and symptoms that continue or develop after acute COVID-19. It includes both ongoing symptomatic COVID-19 (from 4 to 12 weeks) and post-COVID-19 syndrome (12 weeks or more) [1,2,3,4].

We and others have already hypothesized that the onset of neurovegetative symptoms upon SARS-CoV-2 infection, for example, may be caused by the production of autoantibodies through molecular mimicry phenomena [5,6]. Molecular mimicry is probably because, during evolution, proteins improved and/or accumulated functions, modifying their cellular and anatomical localization but sometimes maintaining a strong structural 3D morphology similarity. As a result, peptides sharing identical epitopes may be found in microorganisms (e.g., viruses and bacteria) and in human cells. If present on the cell surface (constitutively or after post-translational modifications due to conditions such as physico-chemical stress or normal aging), these human proteins can be recognized by an activated immune system (both humoral and cellular) as “foreign”, generating an autoimmune response [7,8].

As previously reported by our group, various human proteins that share putatively immunogenic peptides with SARS-CoV-2 proteins have been found. Interestingly, some of them are expressed in the dorsal motor nucleus and nucleus ambiguous [6]. For this reason, we have hypothesized that severe forms of COVID-19 could produce transitory or permanent damages in some vagal structures (i.e., nuclei and ganglia) and, in turn, this can be responsible for a lower vagal tone and all the related clinical signs and symptoms.

In this study we searched the presence of antibodies against two suspected proteins of vagal nuclei—i.e., corticotropin releasing factor receptor 2 and calcitonin gene-related peptide type 1 receptor—using an immunoassay test on blood obtained from patients with cardiorespiratory symptoms after COVID-19 disease (ongoing symptomatic COVID-19 patients, named long-COVID group, according to the NICE guidelines), subjects vaccinated without a history of SARS-CoV-2 infection (vaccinated group), and subjects not vaccinated without a history of SARS-CoV-2 infection (control group). 

The proteins we have chosen are both located in the nucleus ambiguous and share immunogenic peptides with the spike viral protein predicted for B lymphocyte response [6]. 

## 2. Materials and Methods

### 2.1. Patients

Consecutive patients referred to the inpatient pulmonary rehabilitation unit (ICS Maugeri IRCCS, Telese Terme Institute, Telese, Italy) for a 5-week intensive pulmonary rehabilitation program were evaluated for inclusion in the study.

The enrolled individuals included 18 males and 15 females, with an age between 50 and 60 years (Table 1), divided as follows: *i.* 14 long-COVID-19 subjects (1 month post infection by SARS-CoV-2, 13 males and 1 female with a mean age of 60.14 ± 8.97 years); *ii.* 14 vaccinated individuals (1 month post the first dose of the Pfizer-BioNTech COVID-19 vaccine BNT162b2, 3 males and 11 females with a mean age of 54.21 ± 11.83 years), and *iii.* 5 individuals who had not been infected by the novel coronavirus or vaccinated were used as controls (2 males and 3 females with a mean age of 62.4 ± 18.03). All study procedures were performed within 24 h from admission. Such procedures were performed at discharge as well.

### 2.2. Clinical Tests

Spirometry was performed with automated equipment (Vmax^®^ Encore, Vyasis Healthcare, Milan, Italy) and was reported following the most updated ERS/ATS guidelines [9]. Diffused lung capacity for carbon monoxide (DLCO) was performed with the same apparatus. 

Cardiopulmonary exercise testing was performed according to the latest available ERS statement [10] with automated equipment (Vyntus^®^ CPX, CareFusion, San Diego, CA, USA). We used a cycloergometer-based, incremental step protocol. After performing baseline measurements (electrocardiogram, EKG, lung volumes, peripheral oxygen saturation, SpO2, and blood pressure, BP), patients underwent a resting phase of approximately three minutes, then began with an unloaded phase, followed by an incremental step phase. The test was interrupted whenever one of the following occurred: maximal effort reached or early termination due to adverse effects (such as arrhythmias, chest pain).

### 2.3. Statistical Analysis

Statistical analysis was performed with the software SPSS version 29.0 (IBM, Chicago, IL, USA). Continuous variables were reported as mean ± standard deviation in case of normal distributions, otherwise as median (interquartile range, IQR) in case of skewed, non-gaussian distributions. Differences between time-points were then explored using a paired-samples *t*-test in case of normally distributed variables, while the Wilcoxon rank test was employed in case of continuous variables with non-parametric distributions. Categorical variables were expressed as relative frequencies. Relationships between variables were explored by means of Pearson’s and Spearman’s correlations, and predictor factors were explored through linear regression models. Given the small sample size, no sub-analysis was possible.

### 2.4. Biochemical Tests

#### 2.4.1. Sera Samples

Sera samples were centrifuged at 10,000× *g* 4 °C for 10′, and the supernatants were stored at −80 °C for further analysis. A SARS-CoV-2 IgG assay for the quantitative detection of neutralizing antibodies to the SARS-CoV-2 virus was used (Abbott, Sligo, Ireland).

#### 2.4.2. Peptides

The six amino acid-long peptides LVLLPL (named Pcort) and FLVLLP (named Pcalc) from corticotropin-releasing factor receptor 2 (Uniprot ID Q13324) and calcitonin gene-related peptide type 1 receptor (UniProt ID Q16602), respectively, shared with SARS-CoV-2 spike glycoprotein (UniProt ID P0DTC2), were synthetized and purchased from Elabscience Technology Inc. (Houston, TX, USA).

Both peptides were synthetized as longer biotinylated forms, FLVLLPLVSSQCVNL for Pcort and FLVLLPLVSSQCVNL for Pcalc, with a purity ≥99.0% for Pcort and ≥75.0% for Pcalc.

The lyophilized peptides were reconstituted according to the manufacturer’s instructions at a final concentration of 1 mg/mL, and then stored at −20 °C.

#### 2.4.3. Salting Out Technique

To isolate the protein fraction containing IgGs from sera samples, the salting out technique was used. Briefly, sera samples were diluted in TRIS-HCl 1M, pH 7.5, with a final concentration of 1 mg/mL and then precipitated with 25% and 35% saturated ammonium sulfate (NH_4_)_2_SO_4_ 4.1 M, pH 7. The precipitates were dissolved in PBS 1X and stored at −20 °C until use.

#### 2.4.4. Dot-Blot Analysis

The presence of IgGs against the peptides Pcort and Pcalc was tested by performing a dot-blot assay. Briefly, a nitrocellulose membrane (0.2 μm pore size, Amersham™ Protran™, GE HealthCare, Milan, Italy) was pretreated for 10 min with 1.0% glutaraldehyde in PBS 1X. Peptides (5 μg) were spotted on the activated membranes and left to dry at room temperature. The membranes were washed twice in PBS 1Χ and then incubated for 1 h in PBS 1X/0.05% (*v*/*v*) Tween 20 (PBST), containing 5% BSA to block nonspecific binding sites. Following the blocking step, samples (25 ng) were spotted on the membranes and left to dry at room temperature for 1 h. Then, membranes were washed three times in PBST (5′/wash) and incubated for 1 h in PBST/0.5% BSA with peroxidase-conjugated anti-human IgG produced in rabbit (1:10,000, anti-Human IgG, A8792) obtained from Merck KGaA, Darmstadt, Germany. Membranes were washed three times in PBST (5′/wash) and immunoblots were developed using an ECL detection assay (Merck KGaA, Darmstadt, Germany). Signal intensity was quantified using densitometry and shown with increased numbers of plus signs, from negative (−) to strong positivity (++++). 

## 3. Results

### 3.1. Clinical Results

#### Baseline Characteristics of the Study Population

Study sample characteristics, as well as baseline and follow-up assessments, are summarized in Table 2.

Our sample included 14 patients (1 female, 7.1%) with a mean age of 54.7 ± 10.3 years and an elevated BMI (mean 30.92 ± 6.43). At baseline, lung function was slightly impaired, with a mean forced expiratory volume in the first second expressed as a percentage of the predicted values (FEV1%) of 77.9 ± 21.2%, a mean forced vital capacity expressed as percentage of predicted values (FVC%) of 75.2 ± 16.6%, and a mean FEV1/FVC ratio of 83.20 ± 5.61%, thus indicating a possible restrictive syndrome. Gas exchanges were also impaired, with a mean diffusing lung capacity for carbon monoxide expressed as a percentage of the predicted values (DLCO%) of 68.90 ± 20.30%. An important degree of functional impairment was observed at baseline, with a median 6 min walking distance (6MWD) of 280.00 (225.50; 327.25) m, a percent predicted peak consumption of oxygen (VO2 peak %) of 60.50 ± 14.86%, and a percent predicted maximal minute ventilation (VE max %) of 69.10 ± 12.41%.

### 3.2. Biochemical Results

#### Dot-Blot Analysis

Figure 1 shows a representative peptide dot-blot immunoassay analysis in long-COVID, vaccinated, and control groups.

Table 3 precisely details the antigenic pattern of Pcort and Pcalc peptides monitored using sera from 14 long-COVID and 14 vaccinated subjects and from 5 healthy controls. Signal intensity was quantified by densitometry and shown with increase numbers of plus signs, from negative (−) to strong positivity (++++). For both peptides, the reactive pattern with serum from long-COVID-19 patients was like that obtained using the serum from vaccinated subjects (Table 3). In addition, some sera from control donors also reacted by giving signals of minor intensity (+). 

## 4. Discussion

Virus infection is considered a primary factor that has been implicated in the initiation of autoimmune disease. Infection triggers a robust and usually well-coordinated immune response that is critical for viral clearance. However, in some cases, immune regulatory mechanisms may falter, culminating in the breakdown of self-tolerance, resulting in immune-mediated attack directed against both viral and self-antigens [11]. Moreover, vaccines and autoimmunity are linked fields. Vaccine efficacy is based on whether a host immune response against an antigen can elicit a memory T-cell response over time. Although the described side effects thus far have been mostly transient and acute, vaccines are able to elicit the immune system towards an autoimmune reaction [12], but epidemiological studies do not fully support this hypothesis [13].

In this work, we would test if our previous hypothesis, supported by bioinformatic analyses, about the putative autoimmunity generated by SARS-CoV-2 infection against vagal nuclei through molecular mimicry phenomena was supported by biological data or not.

Interestingly, our results show that putative autoantibodies are present in both the long-COVID-19 and vaccinated groups, the latter without ever having shown symptoms of disease, even in a mild form. 

These data led us to postulate the following:As already supposed, the presence of autoantibodies is a necessary but not sufficient condition for the triggering of autoimmunity; indeed, it is necessary that the subjects have predisposing conditions, e.g., physical or chemical stress to their endothelial cells (such as hypertension or diabetes, respectively) [14].It is plausible that the viral load must be very high for the triggering of autoimmunity and vaccination—given that it creates a local reaction that prepares the body to react promptly in the event of viral attack through the upper airways—and prevents a higher viral load as it may happen in unvaccinated people (Figure 2).

What we observed in our experiments is the presence of putative autoantibodies in both the blood of people that have encountered the virus and have contracted the disease and people that have encountered the spike protein through the vaccine, without any possibility to measure the difference in terms of quantity of autoantibodies between subjects in the acute phase of COVID-19 and subjects after vaccination. This issue should be further investigated, if possible, in other studies.

In a previous paper, we postulated that the lower respiratory phase of COVID-19 is a vasculitis of the vascular endothelial cells of the respiratory barrier, putatively induced by a molecular mimicry towards antistress proteins [15]. Based on this study, we can postulate that vaccination—although it could generate putative autoantibodies that, in very unfortunate subjects who are evidently predisposed for genetic or epigenetic reasons, can lead to (fortunately rare) very serious complications—can be decisive in preventing the disease from spreading from a high respiratory phase to the lungs and the rest of the organism; the latter putting the subject not only at risk of life but also at risk of the onset of all those sequelae that nowadays we define as long-COVID. The datum that the presence of autoantibodies is not a sufficient condition to generate autoimmunity is evidence that should stimulate further studies. 

## 5. Conclusions

We are aware that the detection of antibodies against shared antigenic determinants does not confirm alone the involvement of autoimmunity against proteins of vagal nuclei (specifically, corticotropin-releasing factor receptor 2 and calcitonin gene-related peptide type 1 receptor) in the pathogenesis of long-COVID-19 syndrome. It can be supposed that the presence of autoantibodies could be considered as a risk factor for the development of the pathogenic mechanisms of post-acute sequelae of SARS-CoV-2 infection. 

Our work seems to confirm, by in vivo analyses, the previous in silico data of the putative immunoreactive potential of the shared epitopes, but beyond the detection of antibodies, additional studies involving other biological markers are necessary to support the possible development of autoimmune phenomena, whose knowledge may offer further information for the prevention and treatment of the disease. 

## Figures and Tables

**Figure 1 biology-13-00359-f001:**
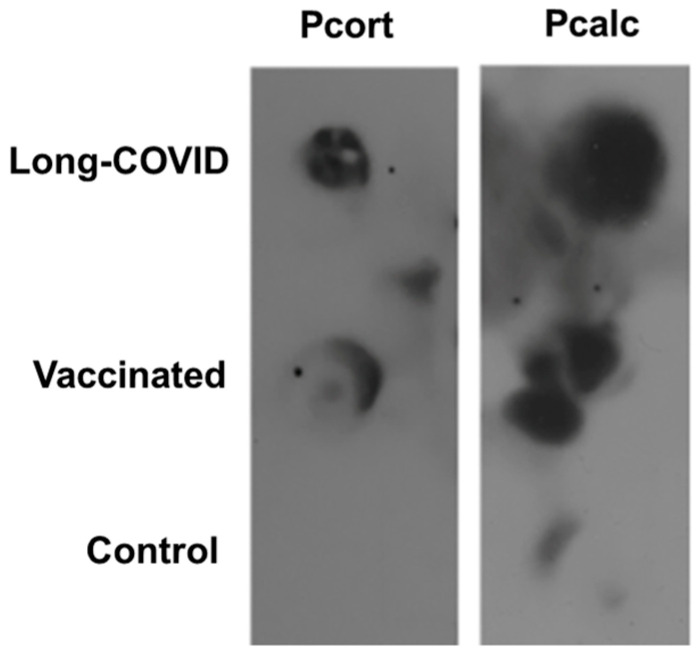
Representative dot-blot immunoassay of the reaction between long-COVID, vaccinated, and healthy control serum with corticotropin-releasing factor receptor 2 (Pcort) and calcitonin gene type 1 receptor (Pcalc)-related peptide. The original dot-blot full image is provided in the Appendix A.

**Figure 2 biology-13-00359-f002:**
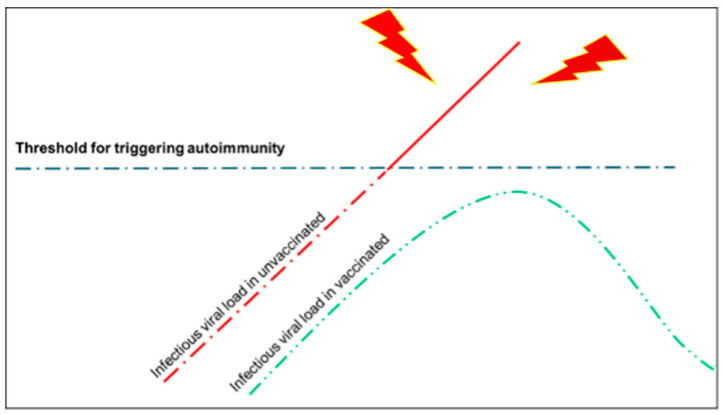
In this draft, we illustrate our hypothesis about the triggering of autoimmunity in subjects exposed to SARS-CoV-2 without having been vaccinated (unvaccinated) or after having been vaccinated, i.e., subjects exposed to spike proteins by vaccination. The latter probably avoids the overcoming of an infective threshold (in terms of viral load) beyond which autoimmunity phenomena may become manifested with clinical signs and symptoms. In other terms, vaccination predisposes the subject to react better to the infection and therefore to contain its spread to other organs other than those of the upper airways (respiratory mucosa, tonsils, etc.).

**Table 1 biology-13-00359-t001:** Patients list.

ID	Sex (%)	Age (M ± SD)	Height (cm)	Weight (Kg)	Smoker	Ex-Smoker	Diabetes	Hypertension	Hypercholesterolemia	IgG
C1	M	57	174	110	No	Yes	No	Yes	No	8.030 S/CO
C2	F	60	162	79	No	No	No	No	No	7.020 S/CO
C3	M	48	177	136	No	Yes	No	No	No	7.890 S/CO
C4	M	71	168	70	No	No	No	Yes	No	5899.10 AU/mL
C5	M	67	172	96	No	No	No	Yes	No	8.690 S/CO
C6	M	54	180	105	No	No	No	No	No	7.800 S/CO
C7	M	58	172	82	No	Yes	No	Yes	No	5.800 S/CO
C8	M	45	170	74	No	No	No	No	No	2.820 S/CO
C9	M	80	168	68	No	Yes	No	No	No	9.240 S/CO
C10	M	62	178	90	No	Yes	No	No	Yes	>80,000.0 AU/mL
C11	M	64	175	90	No	No	No	Yes	No	6.420 S/CO
C12	M	54	178	70	No	No	No	No	No	5.560 S/CO
C13	M	60	191	120	No	Yes	No	No	No	40,060.30 AU/mL
C14	M	62	177	67	Yes	No	Yes	Yes	No	622.90 AU/mL
	7.14% F + 92.86% M	60.14 ± 8.97			7.14% S + 92.86% NS	57.14% NES + 42.86% ES				
V1	F	58	169	75	No	No	No	No	Yes	16,990.70 AU/mL
V2	F	59	165	75	No	No	No	Yes	Yes	14,020.80 AU/mL
V3	F	65	165	50	Yes	No	No	No	No	318.80 AU/mL
V4	F	55	163	93	No	No	No	Yes	Yes	12,411.80 AU/mL
V5	F	56	155	57	No	No	No	No	Yes	8570.80 AU/mL
V6	M	62	170	98	No	No	No	Yes	Yes	11,823.60 AU/mL
V7	M	58	177	76	Yes	No	No	Yes	No	6090.20 AU/mL
V8	F	65	167	57	No	No	No	Yes	Yes	3307.60 AU/mL
V9	F	61	162	54	No	No	No	Yes	Yes	8851.50 AU/mL
V10	F	30	165	60	No	No	No	No	No	17,433.70 AU/mL
V11	M	34	180	102	No	No	No	No	No	41,866.70 AU/mL
V12	F	64	158	75	No	No	No	Yes	Yes	15,048.30 AU/mL
V13	F	36	167	65	No	No	No	No	No	15,645.40 AU/mL
V14	F	56	160	52	No	No	No	Yes	Yes	29,252.30 AU/mL
	78.57% F + 21.43% M	54.21 ± 11.83			14.29% S + 85.71% NS	100% NES + 0% ES				
N1	M	65	170	75	Yes	No	Yes	Yes	No	n.d.
N2	M	58	165	75	No	Yes	No	No	No	9.150 S/CO
N3	F	80	162	85	No	No	No	No	No	n.d.
N4	F	34	163	55	No	No	No	No	No	38.10 AU/mL
N5	F	75	165	80	No	No	No	Yes	No	n.d.
	60% F + 40% M	62.4 ± 18.03			20% S + 80% NS	80% NES + 20% ES				

Abbreviations: AU/mL, arbitrary units per milliliter; C, long-COVID-19; ES, ex-smoker; F, female; IgG, immunoglobulin G; M, male; N, no-COVID/no-vaccine (controls); n.d., no data; NES, no ex-smoker; NS, no smoker; S/CO, signal to cut-off ratio; S, smoker; V, vaccinated. AU/mL: <50 negative; >=50 positive; S/CO: <1.4 negative; >=1.4 positive.

**Table 2 biology-13-00359-t002:** Main demographic, laboratory, lung function, and CPET data at baseline and after pulmonary rehabilitation. Data are presented as mean ± standard deviation or median (interquartile range), unless otherwise specified.

Variable	Values
Subjects, n	14
Females, n (%)	1 (7.1)
Age, years	54.70 ± 10.26
Weight, kg	91.00 ± 21.38
BMI	30.92 ± 6.43
Length of stay, days	27.00 (17.00; 29.00)
HRCT score	12.00 (6.50; 14.00)
HR, bpm	73.90 ± 12.03
SBP, mmHg	125.00 ± 16.33
DBP, mmHg	79.89 ± 5.90
EF, %	60.00 (53.25; 60.00)
GLS, %	−17.46 ± 2.47
FEV_1_, L	2.43 ± 0.52
FEV1%, % predicted	77.9 ± 21.2
FVC, L	2.85 ± 0.63
FVC%, % predicted	75.20 ± 16.59
FEV_1_/FVC	83.20 ± 5.61
DLCO, ml/min/mmHg	18.91 ± 5.66
DLCO, %	68.90 ± 20.30
MIP	93.00 ± 19.60
MIP, %	85.90 ± 15.68
MEP	102.50 ± 23.59
MEP, %	50.80 ± 11.33
6MWD	280.00 (225.50; 327.25)
Fatigue (Borg)	3.00 (3.00; 3.50)
Dyspnea (Borg)	5.00 (3.75; 5.25)
VO_2_ peak	14.96 ± 2.57
VO_2_ peak, %	60.50 ± 14.86
VE/VCO_2_	36.80 ± 4.98
VE/VCO_2_ slope	36.20 ± 3.85
VE max	63.10 ± 12.57
VE max, %	69.10 ± 12.41
PET CO_2_ max, mmHg	31.67 ± 2.59
PET O_2_ max, mmHg	129.09 ± 2.88

Abbreviations: BMI, body mass index; HRCT, high-resolution chest tomography; HR, heart rate; SBP, systolic blood pressure; DBP, diastolic blood pressure; EF, ejection fraction; GLS, global longitudinal strain; FEV1, forced expiratory volume in the first second; FVC, forced vital capacity; DLCO, diffusing capacity of the lungs for carbon monoxide; MIP, maximal inspiratory pressure; MEP, maximal expiratory pressure; 6MWD, 6-min walking distance; VO_2_ peak, peak oxygen uptake; VE/VCO_2_, ratio of minute ventilation to carbon dioxide production; VE max, maximal minute ventilation; PET CO_2_, partial end-tidal pressure of carbon dioxide; PET O_2_, partial end-tidal pressure of oxygen.

**Table 3 biology-13-00359-t003:** Corticotropin-releasing factor receptor 2 peptide (Pcort) and calcitonin gene-related peptide type 1 receptor (Pcalc) antigenic pattern in long-COVID-19 (C1–C14), vaccinated (V1–V14), and control individuals (N1–N5). Dot-blot signal intensity was quantified by densitometry and shown with increased numbers of plus signs such as negative (−), weak (+), moderate (++), strong (+++) and very strong positivity (++++).

Patients	Pcort	Pcalc
Long-COVID		
C1	+++	+++
C2	++++	++
C3	+++	+++
C4	++	+++
C5	+	++
C6	+++	+++
C7	+	+
C8	+	++++
C9	++	+++
C10	++	+
C11	−	+
C12	+++	+
C13	+	−
C14	−	−
Vaccinated		
V1	+	+++
V2	−	++
V3	++	++
V4	+++	+++
V5	++++	++
V6	++	++
V7	++	+
V8	++++	+++
V9	+++	++
V10	+++	+
V11	−	+
V12	++	+
V13	++++	++++
V14	−	++
Controls		
N1	+	+
N2	−	+
N3	−	−
N4	+	−
N5	+	−

## Data Availability

Data are contained within the article and Appendix A. Further inquiries can be directed to the corresponding authors.

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
