# Peer review of "Autoimmunity against Nucleus Ambiguous Is Putatively Possible in Both Long-COVID-19 and Vaccinated Subjects: Scientific Evidence and Working Hypothesis"

_biology, 2024, doi:10.3390/biology13060359_

Round 1

Reviewer 1 Report

Comments and Suggestions for Authors

Title: Autoimmunity against nucleus ambiguous is putatively possible in both post-Covid and vaccinated subjects but vaccine 3 seems to protect from long-Covid syndrome: scientific evidences and working hypothesis.

Manuscript ID: biology-2985859

Article Type: Brief Report

1. Recommendation

Minor Revision

2. Comments to Author an Editor:

The presence of autoantibodies during and immediately after recovery from a viral infection episode is a common phenomenon. Generally, these antibodies disappear relatively quickly during the weeks following the onset of the infection. However, in some viral infections (for example, by Epstein-Barr virus and by citomegalovirus) the titles of autoantibodies may persist raised and associated with reumatological manifestations.

The infection by SARS-CoV-2 is also associated with the production of autoantibodies during the acute phase of COVID-19 (it contributes to the pathological events that characterize this phase of the disease). These autoantibodies, as it occurs in other viral infections, generally disappear from the circulation during the weeks following the infectious episode. Nevertheless, some studies have demonstrated the persistence of autoantibodies in individuals who suffer from post Covid syndrome. In fact, it has been reported cases of autoimmune diseases after COVID-19 in patients without prior autoimmunity; among them, arthritis, lupus erythematous systemic and myositis.

Having said that, it has been a pleasure to revise the manuscript “Autoimmunity against nucleus ambiguous is putatively possible in both post-Covid and vaccinated subjects but vaccine 3 seems to protect from long-Covid syndrome: scientific evidences and working hypothesis”, submitted for publication in Biology. In general, I consider that we are in the presence of a useful and interesting work that, after some necessary changes, must be published .

Despite of the good quality of this study, I have some questions, doubts and suggestions for the authors. I am going to mention those questions, doubts and suggestions in the same order they appear in the manuscript:

I- This manuscript is the logical continuation of a previous paper by the authors (Marino Gammazza, A.; Légaré, S.; Lo Bosco, G.; Fucarino, A.; Angileri, F.; Oliveri, M., Cappello, F. Molecular mimicry in the 311 post-COVID-19 signs and symptoms of neurovegetative disorders? Lancet Microbe 2021, 2(3), e94, doi: 10.1016/S2666-312 5247(21)00033-1). In this manuscript, the authors investigated the presence of autoantibodies against two proteins of vagal nuclei sharing a peptide with SARS-CoV-2 Spike glycoprotein by an immunoassay test on blood obtained from patients with cardiorespiratory symptoms after COVID-19 disease, subjects vaccinated without a history of SARS-CoV-2 infection and subjects not vaccinated without a history of SARS-CoV-2 infection. The demonstration of those antibodies is a very interesting result and its publication could be very useful. However, from the title, the authors are excessively speculative. The detection of antibodies against shared antigenic determinants, without the existence of other elements, is still far from demonstrating the presence of autoimmune phenomena. Added to this is that, beyond the detection of antibodies, additional studies were not carried out in the three groups, which did not contribute to supporting the possible development of autoimmune phenomena in the patient group.

II- Between lines 87 and 92 the authors write “To test our hypothesis, in this study we searched the presence of antibodies against two suspected proteins of vagal nuclei – i.e., corticotropin releasing factor receptor 2 and cal-citonin gene related peptide type 1 receptor – by an immunoassay test on blood obtained from patients with cardiorespiratory symptoms after COVID-19 disease (post-COVID group), subjects vaccinated without a history of SARS-CoV-2 infection (vaccinated group) and subjects not vaccinated without a history of SARS-CoV-2 infection (control group)”. The mere detection of those antibodies, as the authors comment later, cannot prove the hypothesis of the autoimmune phenomena referred to. I suggest the authors rewrite that fragment, removing the phrase “To test our hypothesis.”

III- Between lines 237 and 239 the authors write “…, vaccines are able to elicit the immune system towards an autoimmune reaction [11], but epidemiological studies do not full support this hypothesis [12]. This statement is correct, it is supported by the results of other authors. This being so, and with all the participating individuals converging in the same hospital center, why were the rest of the clinical tests not carried out on the individuals in the vaccinated and control groups? The normality of the remaining examinations in the vaccinated and control groups would have been an additional element in favor of the role of the antibodies detected in autoimmune phenomena in post-Covid patients.

IV- Between lines 283 and 286 the authors write “Finally, our work seems to confirm, by in vivo analyses, the previous in silico data on the involvement of autoimmunity against proteins of vagal nuclei (specifically, corticotropin releasing factor receptor 2 and calcitonin gene related peptide type 1 receptor) in the pathogenesis of long-covid” This conclusion is too categorical and must be softened because the finding of antibodies that react with antigenic determinants in the host and in the virus does not confirm “the involvement of autoimmunity against proteins of vagal nuclei (specifically, corticotropin releasing factor receptor 2 and calcitonin gene related peptide type 1 receptor) in the pathogenesis of long-covid” as the authors write.

Comments on the Quality of English Language

Title: Autoimmunity against nucleus ambiguous is putatively possible in both post-Covid and vaccinated subjects but vaccine 3 seems to protect from long-Covid syndrome: scientific evidences and working hypothesis.

Manuscript ID: biology-2985859

Article Type: Brief Report

1. Recommendation

Minor Revision

2. Comments to Author an Editor:

The presence of autoantibodies during and immediately after recovery from a viral infection episode is a common phenomenon. Generally, these antibodies disappear relatively quickly during the weeks following the onset of the infection. However, in some viral infections (for example, by Epstein-Barr virus and by citomegalovirus) the titles of autoantibodies may persist raised and associated with reumatological manifestations.

The infection by SARS-CoV-2 is also associated with the production of autoantibodies during the acute phase of COVID-19 (it contributes to the pathological events that characterize this phase of the disease). These autoantibodies, as it occurs in other viral infections, generally disappear from the circulation during the weeks following the infectious episode. Nevertheless, some studies have demonstrated the persistence of autoantibodies in individuals who suffer from post Covid syndrome. In fact, it has been reported cases of autoimmune diseases after COVID-19 in patients without prior autoimmunity; among them, arthritis, lupus erythematous systemic and myositis.

Having said that, it has been a pleasure to revise the manuscript “Autoimmunity against nucleus ambiguous is putatively possible in both post-Covid and vaccinated subjects but vaccine 3 seems to protect from long-Covid syndrome: scientific evidences and working hypothesis”, submitted for publication in Biology. In general, I consider that we are in the presence of a useful and interesting work that, after some necessary changes, must be published .

Despite of the good quality of this study, I have some questions, doubts and suggestions for the authors. I am going to mention those questions, doubts and suggestions in the same order they appear in the manuscript:

I- This manuscript is the logical continuation of a previous paper by the authors (Marino Gammazza, A.; Légaré, S.; Lo Bosco, G.; Fucarino, A.; Angileri, F.; Oliveri, M., Cappello, F. Molecular mimicry in the 311 post-COVID-19 signs and symptoms of neurovegetative disorders? Lancet Microbe 2021, 2(3), e94, doi: 10.1016/S2666-312 5247(21)00033-1). In this manuscript, the authors investigated the presence of autoantibodies against two proteins of vagal nuclei sharing a peptide with SARS-CoV-2 Spike glycoprotein by an immunoassay test on blood obtained from patients with cardiorespiratory symptoms after COVID-19 disease, subjects vaccinated without a history of SARS-CoV-2 infection and subjects not vaccinated without a history of SARS-CoV-2 infection. The demonstration of those antibodies is a very interesting result and its publication could be very useful. However, from the title, the authors are excessively speculative. The detection of antibodies against shared antigenic determinants, without the existence of other elements, is still far from demonstrating the presence of autoimmune phenomena. Added to this is that, beyond the detection of antibodies, additional studies were not carried out in the three groups, which did not contribute to supporting the possible development of autoimmune phenomena in the patient group.

II- Between lines 87 and 92 the authors write “To test our hypothesis, in this study we searched the presence of antibodies against two suspected proteins of vagal nuclei – i.e., corticotropin releasing factor receptor 2 and cal-citonin gene related peptide type 1 receptor – by an immunoassay test on blood obtained from patients with cardiorespiratory symptoms after COVID-19 disease (post-COVID group), subjects vaccinated without a history of SARS-CoV-2 infection (vaccinated group) and subjects not vaccinated without a history of SARS-CoV-2 infection (control group)”. The mere detection of those antibodies, as the authors comment later, cannot prove the hypothesis of the autoimmune phenomena referred to. I suggest the authors rewrite that fragment, removing the phrase “To test our hypothesis.”

III- Between lines 237 and 239 the authors write “…, vaccines are able to elicit the immune system towards an autoimmune reaction [11], but epidemiological studies do not full support this hypothesis [12]. This statement is correct, it is supported by the results of other authors. This being so, and with all the participating individuals converging in the same hospital center, why were the rest of the clinical tests not carried out on the individuals in the vaccinated and control groups? The normality of the remaining examinations in the vaccinated and control groups would have been an additional element in favor of the role of the antibodies detected in autoimmune phenomena in post-Covid patients.

IV- Between lines 283 and 286 the authors write “Finally, our work seems to confirm, by in vivo analyses, the previous in silico data on the involvement of autoimmunity against proteins of vagal nuclei (specifically, corticotropin releasing factor receptor 2 and calcitonin gene related peptide type 1 receptor) in the pathogenesis of long-covid” This conclusion is too categorical and must be softened because the finding of antibodies that react with antigenic determinants in the host and in the virus does not confirm “the involvement of autoimmunity against proteins of vagal nuclei (specifically, corticotropin releasing factor receptor 2 and calcitonin gene related peptide type 1 receptor) in the pathogenesis of long-covid” as the authors write.

Author Response

Thank you for giving us the opportunity to revise our manuscript. 

Reviewer 1 comment #1 (R1C#1): Minor Revision

The presence of autoantibodies during and immediately after recovery from a viral infection episode is a common phenomenon. Generally, these antibodies disappear relatively quickly during the weeks following the onset of the infection. However, in some viral infections (for example, by Epstein-Barr virus and by citomegalovirus) the titles of autoantibodies may persist raised and associated with reumatological manifestations. The infection by SARS-CoV-2 is also associated with the production of autoantibodies during the acute phase of COVID-19 (it contributes to the pathological events that characterize this phase of the disease). These autoantibodies, as it occurs in other viral infections, generally disappear from the circulation during the weeks following the infectious episode. Nevertheless, some studies have demonstrated the persistence of autoantibodies in individuals who suffer from post Covid syndrome. In fact, it has been reported cases of autoimmune diseases after COVID-19 in patients without prior autoimmunity; among them, arthritis, lupus erythematous systemic and myositis. Having said that, it has been a pleasure to revise the manuscript “Autoimmunity against nucleus ambiguous is putatively possible in both post-Covid and vaccinated subjects but vaccine 3 seems to protect from long-Covid syndrome: scientific evidences and working hypothesis”, submitted for publication in Biology. In general, I consider that we are in the presence of a useful and interesting work that, after some necessary changes, must be published. Despite of the good quality of this study, I have some questions, doubts and suggestions for the authors.

Authors’ Reply (AR): Thank you very much for this positive comment.

R1C#2: I am going to mention those questions, doubts and suggestions in the same order they appear in the manuscript:

I- This manuscript is the logical continuation of a previous paper by the authors (Marino Gammazza, A.; Légaré, S.; Lo Bosco, G.; Fucarino, A.; Angileri, F.; Oliveri, M., Cappello, F. Molecular mimicry in the 311 post-COVID-19 signs and symptoms of neurovegetative disorders? Lancet Microbe 2021, 2(3), e94, doi: 10.1016/S2666-312 5247(21)00033-1). In this manuscript, the authors investigated the presence of autoantibodies against two proteins of vagal nuclei sharing a peptide with SARS-CoV-2 Spike glycoprotein by an immunoassay test on blood obtained from patients with cardiorespiratory symptoms after COVID-19 disease, subjects vaccinated without a history of SARS-CoV-2 infection and subjects not vaccinated without a history of SARS-CoV-2 infection. The demonstration of those antibodies is a very interesting result and its publication could be very useful. However, from the title, the authors are excessively speculative. The detection of antibodies against shared antigenic determinants, without the existence of other elements, is still far from demonstrating the presence of autoimmune phenomena. Added to this is that, beyond the detection of antibodies, additional studies were not carried out in the three groups, which did not contribute to supporting the possible development of autoimmune phenomena in the patient group.

AR: Thank you again for the positive comment. We modified the title and other sentences throughout the text (see the conclusions section) moderating the speculative aspects of the work. 

II- Between lines 87 and 92 the authors write “To test our hypothesis, in this study we searched the presence of antibodies against two suspected proteins of vagal nuclei – i.e., corticotropin releasing factor receptor 2 and calcitonin gene related peptide type 1 receptor – by an immunoassay test on blood obtained from patients with cardiorespiratory symptoms after COVID-19 disease (post-COVID group), subjects vaccinated without a history of SARS-CoV-2 infection (vaccinated group) and subjects not vaccinated without a history of SARS-CoV-2 infection (control group)”. The mere detection of those antibodies, as the authors comment later, cannot prove the hypothesis of the autoimmunephenomena referred to. I suggest the authors rewrite that fragment, removing the phrase “To test our hypothesis.”

AR: We modified the text according to the request.

III- Between lines 237 and 239 the authors write “…, vaccines are able to elicit the immune system towards an autoimmune reaction [11], but epidemiological studies do not full support this hypothesis [12]. This statement is correct, it is supported by the results of other authors. This being so, and with all the participating individuals converging in the same hospital center, why were the rest of the clinical tests not carried out on the individuals in the vaccinated and control groups? The normality of the remaining examinations in the vaccinated and control groups would have been an additional element in favor of the role of the antibodies detected in autoimmune phenomena in post-Covid patients.

AR: The vaccinated and healthy control experienced no symptoms. Furthermore, the ethical committee had not allowed to perform clinical examination to control subjects.

IV- Between lines 283 and 286 the authors write “Finally, our work seems to confirm, by in vivo analyses, the previous in silico data on the involvement of autoimmunity against proteins of vagal nuclei (specifically, corticotropin releasing factor receptor 2 and calcitonin gene related peptide type 1 receptor) in the pathogenesis of long-covid” This conclusion is too categorical and must be softened because the finding of antibodies that react with antigenic determinants in the host and in the virus does not confirm “the involvement of autoimmunity against proteins of vagal nuclei (specifically, corticotropin releasing factor receptor 2 and calcitonin gene related peptide type 1 receptor) in the pathogenesis of long-covid” as the authors write.

AR: Thank you for this comments, we modified the conclusion.

Reviewer 2 Report

Comments and Suggestions for Authors

The authors demonstrate the presence of IgG antibodies against Pcort and Pcalc in patients with Post-COVID symptoms and people vaccinated against COVID-19. The control group contains unvaccinated people who never suffered from COVID-19. 

They explain the relevance of the examined antibodies as they show peptide resemblance with Spike glycoprotein. The according proteins are found in the vagal nuclei which might explain postural orthostatic  symptoms linked to vagal reactions. 

My detailed comments are as followed: 

In general, I woul recommend to proper distinguish between Long-COVID and Post-COVID in the introduction (eg. As proposed in the according NICE Guidelines (www.nice.org.uk/guidance/ng188 last updated 25 january 2024 and the WHO clinical case definitioni of WHO dephi consensos 6 october 2021)

Study design: 

·         Subjects vaccinated without a history of SARS-CoV-2 infection and subjects not vaccinated without a history of SARS-CoV-2 infection: Ì would recommend to provide proof, that these subjects never had a silent SARS-CoV-2 infection (eg. with serology of SARS-CoV2 Spike and nucleocapsid antibodies?)

·         Patients with cardiorespiratory symptoms after COVID-19 disease: From the clinicians point of view, it would be very necessary to provide further details (eg. Severity of disease like hospitalised or not or „had to stay in bed“ – time and antiviral treatment yes or no, time of infection (referring to the variant of SARS-CoV2 would be of interest) and detailled list of symptoms and to describe the way of diagnosis of Long-COVID was made (which means to rule out some differential diagnoses). According to table 2 I assume that a detailled clinical examination took place, yet the data should be available and could be additionally provided in a supplement table. Did all patients have proven COVID-infection with PCR – test?

·         Did the subjects not vaccinated without a history of SARS-CoV-2 infection suffer from any postVACC – symptoms or were they a healthy control?

·         Detailed description of Spirometry and spyroergometer-test is not necessary in my point of view.

·         Dot blot analysis is only a semiqualitative analysis. For further insights (eg. Comparing the patients who suffered from SARS-CoV-2 infection and the „uninfected vaccinated“ people, it would be very interessting to have a quantification of antibody levels.

·         I would appreciate another controll group: people, who had proven COVID-infection but no signs or symptoms of Post-COVID. Do they also have these antibodies?

Discussion:

·         I cannot follow the postulation Nr. 2 in consequence of the presented study as there is no referrence to viral load in the experiments. In my opinion, this is an „educated guess“ but not a consequence of the provided data.

·         Starting from line 269-281, I comment, that the current study is about Post-COVID so the detailed phases of acute COVID-19 disease are irrelevant for the topic 

Author Response

Thank you for giving us the opportunity to revise our manuscript

Reviewer 2 comment #1 (R2C#1):

The authors demonstrate the presence of IgG antibodies against Pcort and Pcalc in patients with Post-COVID symptoms and people vaccinated against COVID-19. The control group contains unvaccinated people who never suffered from COVID-19. They explain the relevance of the examined antibodies as they show peptide resemblance with Spike glycoprotein. The according proteins are found in the vagal nuclei which might explain postural orthostatic symptoms linked to vagal reactions. My detailed comments are as followed: In general, I woul recommend to proper distinguish between Long-COVID and Post-COVID in the introduction (eg. As proposed in the according NICE Guidelines (www.nice.org.uk/guidance/ng188 last updated 25 january 2024 and the WHO clinical case definitioni of WHO dephi consensos 6 october 2021).

Authors’ Reply (AR): Thank you for this comment. Our patients were discharged within 30 days from an acute setting with signs and symptoms of the disease from 4 weeks up to 12 weeks. Following the NICE guidelines, our patients can be classified as long covid - ongoing symptomatic COVID-19 (signs and symptoms of the disease from 4 weeks up to 12 weeks) and we named this group long-covid. For the benefit of clarity, we modified the text accordingly.

      R2C#2: Study design: Subjects vaccinated without a history of SARS-CoV-2 infection and subjects not vaccinated without a history of SARS-CoV-2 infection: Ì would recommend to provide proof, that these subjects never had a silent SARS-CoV-2 infection (eg. with serology of SARS-CoV2 Spike and nucleocapsid antibodies?).

 AR: Unfortunately we do not have these informations. All controls had never experienced SARS-CoV-2 infection symptoms.

R2C#3: Patients with cardiorespiratory symptoms after COVID-19 disease: From the clinicians point of view, it would be very necessary to provide further details (eg. Severity of disease like hospitalised or not or „had to stay in bed“– time and antiviral treatment yes or no, time of infection (referring to the variant of SARS-CoV2 would be of interest) and detailled list of symptoms and to describe the way of diagnosis of Long-COVID was made (which means to rule out some differential diagnoses). According to table 2 I assume that a detailled clinical examination took place, yet the data should be available and could be additionally provided in a supplement table. Did all patients have proven COVID-infection with PCR – test?

 AR: All patients had been previously hospitalized and had been admitted to intensive rehabilitation Unit within 30 days of discharge for the persistence of symptoms. No one had antiviral treatment. The patient included in the study were long Covid patient as specified in the text. In all patients the diagnosis of SARS COV2 –infection had been confirmed by PCR – test.

 R2C#4: Did the subjects not vaccinated without a history of SARS-CoV-2 infection suffer from any postVACC – symptoms or were they a healthy control?

 AR: No any post vacc symptoms. they were healthy control.

 R2C#5. Detailed description of Spirometry and spyroergometer-test is not necessary in my point of view.

 AR: We modified the text.

 R2C#6: Dot blot analysis is only a semiqualitative analysis. For further insights (eg. Comparing the patients who suffered from SARS-CoV-2 infection and the, uninfected vaccinated“ people, it would be very interesting to have a quantification of antibody levels. I would appreciate another control group: people, who had proven COVID-infection but no signs or symptoms of Post-COVID. Do they also have these antibodies?

 AR: Unfortunately, we do not have informations regarding the antibody levels and we have not the opportunity to perform this test. So, what we observed in our experiments, as already reported in the manuscript, is the presence of putative autoantibodies in both the blood of people that have encountered the virus and have done the disease (long-COVID) and people that have encountered the spike protein through the vaccine, without any possibility to measure the difference in terms of quantity of autoantibodies between subjects in acute phase of COVID-19 and subjects after vaccination. This issue should be further investigated, is possible, in other studies. Moreover, we do not have another control group: people, who had proven COVID-infection but no signs or symptoms of long-COVID; it would be certainly interesting to determine if they have these antibodies, but this means another study and another recruitment of patients. We hope to have the opportunity to do this in the future.

 R2C#7: Discussion: I cannot follow the postulation Nr. 2 in consequence of the presented study as there is no reference to viral load in the experiments. In my opinion, this is an „educated guess“ but not a consequence of the provided data.

AR: We modified the text.

R2C#8: Starting from line 269-281, I comment, that the current study is about long-COVID so the detailed phases of acute COVID-19 disease are irrelevant for the topic.

AR: We modified the text

Reviewer 3 Report

Comments and Suggestions for Authors

This manuscript described the causative mechanisms of long-COVID, and autoimmunity and the production of autoantibodies targeting self-antigens via the molecular mimicry phenomenon seem to have a role. The authors evaluated the presence of autoantibodies against two proteins of vagal nuclei sharing a peptide with SARS-CoV-2 Spike glycoprotein in sera, and the possible antibodies against two suspected proteins of vagal nuclei – i.e., corticotropin releasing factor receptor 2 and calcitonin gene related peptide type 1 receptor.

In general, this is a well-written manuscript. Other points in this manuscript needed to be clarified are listed below:

In general: please unify the term, long-COVID or long COVID-19 (page 2, graphical abstract)?

Materials and Methods:

1.     This article is a cross sectional study, and some important data were missing, such as the duration of long-COVID and vaccination of COVID-19. Besides, SARS-CoV-2 Spike glycoprotein IgG and inflammatory makers- i.e., IL6 are also informative to support the authors’ hypothesis.     

2.     I am curious about the outcome in the population of COVID-19 infected patients without long-COVID. Would the result be different or similar? Could you provide the data?

3.     Do you have serial data of LVLLPL (named Pcort) and FLVLLP (named Pcalc) among long-COVID patients?

Results:

1.     Page 7: the data is duplicated “MEP, % 50.80±11.33”

Author Response

Thank you for giving us the opportunity to revise our manuscript

Reviewer 3 comment #1 (R3C#1): This manuscript described the causative mechanisms of long-COVID, and autoimmunity and the production of autoantibodies targeting self-antigens via the molecular mimicry phenomenon seem to have a role. The authors evaluated the presence of autoantibodies against two proteins of vagal nuclei sharing a peptide with SARS-CoV-2 Spike glycoprotein in sera, and the possible antibodies against two suspected proteins of vagal nuclei – i.e., corticotropin releasing factor receptor 2 and calcitonin gene related peptide type 1 receptor. In general, this is a well-written manuscript.

Authors’ Reply (AR): Thank you for this positive comment.

R3C#2: Other points in this manuscript needed to be clarified are listed below: In general: please unify the term, long-COVID or long COVID-19 (page 2, graphical abstract)?

AR: we modified the text.

R3C#3: Materials and Methods: This article is a cross sectional study, and some important data were missing, such as the duration of long-COVID and vaccination of COVID-19. Besides, SARS-CoV-2. Spike glycoprotein IgG and inflammatory makers- i.e., IL6 are also informative to support the authors’ hypothesis. 

AR: Our patients were long-covid patients (ongoing symptomatic COVID-19). They were discharged within 30 days from an acute setting.  The vaccinated group included serum sample collected 1 month post the first dose of the Pfizer-BioNTech COVID-19 vaccine BNT162b2. We added in table 1 the results for SARS-CoV-2 IgG assay. Regarding inflammatory marker like IL-6 we do not have this data. We are aware that there are many molecules to be investigated, informative to support our hypothesis. We would like to continue this project and this issue should be further investigated, if possible, in other studies. then we hope to go deep in a further work.

R3C#4: I am curious about the outcome in the population of COVID-19 infected patients without long-COVID. Would the result be different or similar? Could you provide the data?

AR: Our patients were long-covid patients (ongoing symptomatic COVID-19), we have not data regarding acute COVID-19.

R3C#5: Do you have serial data of LVLLPL (named Pcort) and FLVLLP (named Pcalc) among long-COVID patients?

AR: We do not have these data.

R3C#6: Results: Page 7: the data is duplicated “MEP, % 50.80±11.33”

AR: We modified the text.

Round 2

Reviewer 3 Report

Comments and Suggestions for Authors

Dear authors:

Thank for your revision and reply, and I have no more further question. 

Best regards!